# Enhancing Obstetric Healthcare Providers’ Knowledge of Black Maternal Mental Health: A Feasibility Study

**DOI:** 10.3390/ijerph21101374

**Published:** 2024-10-17

**Authors:** Kortney Floyd James, Keisha Reaves, Misty C. Richards, Kristen R. Choi

**Affiliations:** 1School of Nursing, University of California Los Angeles, Los Angeles, CA 90095, USA; krchoi@ucla.edu; 2RAND Corporation, Santa Monica, CA 90401, USA; 3PushThru Therapy, Atlanta, GA 30338, USA; workoutcouch@gmail.com; 4School of Medicine, University of California Los Angeles, Los Angeles, CA 90095, USA; mcrichards@mednet.ucla.edu

**Keywords:** nursing education, perinatal mental health, cultural awareness, clinical competence

## Abstract

Despite guidelines for screening and treating perinatal mood and anxiety disorders (PMADs), systemic issues and clinician biases often result in unmet mental health needs in Black women. This study assessed the feasibility and impact of comprehensive PMAD training on obstetric healthcare providers’ attitudes, knowledge, and implicit racial biases. We conducted a feasibility study with two cohorts of healthcare providers who received either in-person or virtual training. The training focused on PMADs, implicit bias, and culturally responsive care. Participants completed pre- and post-training assessments measuring attitudes, knowledge, empathy, and implicit racial biases. Both training modalities showed trends towards improved PMAD screening attitudes and empathy, with significant increases in beliefs about treatment efficacy. Implicit bias scores approached significance, showing a shift toward fewer participants with racial preferences. However, there was an unexplained increase in preference for White over Black post training. The training improved healthcare providers’ readiness to screen for PMADs and enhanced their understanding of PMADs. However, the persistence of implicit biases highlights the need for ongoing, sustained interventions to address deeply rooted biases. Future research should incorporate continuous learning strategies and link training to healthcare outcomes for minoritized communities.

## 1. Introduction

Black women comprise 9% of the population in Los Angeles County, California, but have the highest prevalence of symptoms of depression and anxiety (38%) during pregnancy and postpartum compared to women of other racial groups [1]. While individual-level factors such as stigma and provider mistrust contribute to untreated depression and anxiety [2,3], the role of clinicians in perpetuating these disparities cannot be ignored. Despite receiving care within healthcare systems, 69.1% to 86.5% of Black women with moderate-to-severe depression do not receive mental health treatment [4,5], putting them at increased risk for premature birth, low-birth-weight infants, impaired bonding, and suicide [6,7,8]. Black women living in Los Angeles County, as well as other major cities in California (e.g., San Francisco), reported that their symptoms of depression and/or anxiety were dismissed by their obstetric provider or misdiagnosed as side effects from medications [9]. This neglect may reflect underlying obstetric racism [10,11,12] or other forms of discrimination (e.g., insurance/class, language, sexual orientation) [13,14,15,16], which further erode women’s trust in healthcare providers and hinder them from disclosing symptoms of perinatal mood and anxiety disorders (PMADs). Consequently, untreated PMAD symptoms often escalate, leading to emergency-care visits and short-term hospital admissions that do not address long-term mental health needs [17,18,19,20,21].

Although guidelines from the U.S. Preventive Services Task Force [22], American College of Obstetricians and Gynecologists [23], and American College of Nurse-Midwives [24] exist to ensure appropriate treatment of PMADs, many clinicians, specifically physicians and nurse practitioners, fail to follow these due to time constraints, lack of knowledge, and insufficient training [25]. Of the few studies that have attempted to address PMAD screening and referral disparities, most aimed to improve physician, midwife, or nurse practitioner screening practices [25,26,27]. These studies lacked reflective practices for clinicians to address their biases and inclusive education to provide culturally relevant care to minoritized or marginalized groups (e.g., racial and ethnic minoritized groups and gender-expansive people). Furthermore, these studies did not include nurses or medical assistants, who are often the first healthcare providers that patients interact with and are responsible for conducting initial screenings for PMADs, which can set the tone for the visit [28]. As the “first line” in patient care, it is essential that nurses and medical assistants are knowledgeable and receive training on PMADs and administering mental health screenings.

Comprehensive PMAD training is necessary to enhance all healthcare providers’ (i.e., medical assistants, nurses, physicians) ability to address the racial and cultural factors influencing Black women’s mental health during pregnancy and postpartum. Such training should not only cover information about various PMADs, including signs, symptoms, treatment, and screening, but also provide culturally responsive guidance that addresses the unique needs of Black women and birthing people, as well as the barriers they encounter within healthcare systems and everyday life related to discrimination, bias, and racism [29]. Thus, we conducted a feasibility study to evaluate the impact of comprehensive PMAD training (Black Maternal Mental Health 101) on obstetric healthcare providers’ (1) attitudes towards patients with PMADs, (2) knowledge about PMADs, (3) readiness to assess PMAD symptoms, (4) level of empathy, and (5) implicit racial biases. We examined pre-/post-test changes to these measures and compared two training modalities, virtual and in-person.

## 2. Materials and Methods

We conducted an educational feasibility study with two separate cohorts of healthcare providers. The first cohort attended the 2-day in-person Black Maternal Mental Health Training (BMMH 101), while the second cohort attended virtually (Figure 1).

### 2.1. Conceptual Model

The content of the BMMH 101 training, including data collection, analysis, and interpretation of results, was guided by the principles of Reproductive Justice (RJ). The RJ movement, founded by Black feminists in 1994 [30], underscores the right of Black women and birthing people to raise their children in a safe and healthy environment, free from systemic discrimination. RJ serves as a key framework for this study by addressing the intersection of reproductive rights and social justice in the context of perinatal mental health. RJ emphasizes the need for equitable access to reproductive health services while recognizing how intersecting systems of discrimination—such as racism, classism, and sexism—impact marginalized and minoritized communities. This framework advocates for a holistic approach to reproductive health that not only includes access to health services but also addresses the broader social conditions that affect individuals’ ability to exercise their reproductive rights fully and freely. In this study, the RJ framework guided the development and implementation of the BMMH 101 training to ensure that the training provided a comprehensive lens for clinicians to understand and address the unique needs of Black women.

### 2.2. Intervention

The BMMH 101 training was developed by Keisha Reaves, a Licensed Professional Counselor with over 16 years of experience, who is certified in counseling Black women throughout the reproductive cycle (e.g., fertility, conception, infant loss, postpartum). The BMMH 101 training has been provided to various healthcare providers by Keisha Reaves and consists of two didactic lectures (eight hours each, over 2 days) with interactive roleplay (Table 1). The training was designed to enhance healthcare providers’ cultural awareness, enabling them to better understand Black women’s experiences and perspectives and build rapport with their Black patients and families. The 2-day in-person training occurred over a weekend at a healthcare facility in Los Angeles County in March 2024. For the virtual training, a remotely located instructor, (redacted for anonymity), provided participants with real-time guidance via Zoom; this 2-day training also occurred over a weekend in March 2024.

### 2.3. Recruitment

The research team collaborated with office and nurse managers at various outpatient obstetric care offices and birthing centers in Los Angeles County, California, to plan and facilitate recruitment for the feasibility study. During these discussions, it became evident that many of the outpatient obstetric care offices/clinics had a disproportionate staffing structure. Some facilities employed only one or two registered nurses (RNs) who rotated between multiple locations, while there were typically six to nine medical assistants (MAs) who predominantly remained at a single site. Given this staffing imbalance, the recruitment criteria were expanded from focusing solely on RNs to include all healthcare workers in outpatient obstetric care offices and inpatient birthing centers. This adjustment ensured a broader and more representative sample of healthcare providers in these settings. Managerial staff at each site sent emails to all staff who were eligible. The email contained digital flyers with information about the BMMH 101 training and a hyperlink to register for the in-person or virtual training. Eligibility criteria were the same across all sites: (1) employees aged 18 or older; (2) active license as a clinician (e.g., midwife, nurse, physician, social worker) or completed training and certification (e.g., doula, medical assistant) in California; (3) involved with the care of pregnant and/or postpartum patients at their respective site. Figure 2 presents a study flow diagram of participant recruitment and participation. The feasibility study protocol was approved by the Institutional Review Board (IRB) of the Human Research Protections Office of the sponsoring institution (#23-000499).

### 2.4. Data Collection

Participants in both cohorts completed an online Qualtrics pre-test in their location of choice, outside of the training room and not during the training session, after reviewing the study information. The pre-test included nine demographic questions on age, racial identity, education, work history, and training goals. The pre-test also assessed participants’ baseline attitudes towards PMADs, knowledge of PMADs, readiness to assess PMAD symptoms, level of empathy, and implicit racial biases. Table 2. Four weeks after completing the BMMH 101 training, participants who attended both days took a post-test survey to measure the BMMH 101 training’s impact on these areas and give participants the opportunity to provide feedback on what went well and what could be improved.

### 2.5. Data Analysis

Data from both cohorts were analyzed using R, version 4.3.1. We used frequencies and descriptive statistics to examine characteristics of the study sample. We calculated means, standard deviations, ranges, and histograms for each of the outcome variables and their subscales to examine distribution of scores in the sample. Then, paired sample t-tests were used to compare pre- and post-education differences in PASAQ and JSE overall scores and subscale scores. For the IAT, participants were categorized into those who had no implicit racial preference, those with a White preference, and those with a Black preference based on their scores from the Affect Misattribution Procedure. Pre- and post-test differences in IAT group were examined using a Pearson chi-square test. Only participants with both pre- and post-test measures were included in these analyses. We also used independent sample *t*-tests to compare differences in pre-test to post-test score changes for in-person versus virtual participation. *p*-values of less than 0.05 were considered statistically significant.

## 3. Results

There were 22 participants in the sample, 6 of whom participated in person and 16 of whom participated virtually. On average, participants were 39.7 years of age (SD = 12.3, range = 26–61 years), and 100% of participants identified as women. Forty percent (*n* = 9) identified their race as Black, 31.8% (*n* = 7) as White, and 9.1% (*n* = 2) as East or Southeast Asian; 27.3% (*n* = 5) identified as Hispanic and/or Latina. Participants were registered nurses (RNs, *n* = 12, 54.5%), medical assistants (*n* = 4, 18.2%), mental health specialists (*n* = 4, 18.2%), and a certified professional midwife (*n* = 1, 4.5%). On average, participants had 24.9 years of experience (SD = 10.9, range = 8 months–38 years).

Mean pre- and post-test differences in outcome measures for all scales and subscales are shown in Table 3. Overall, the sample had relatively high levels of PMAD screening acceptability and empathy at baseline. Scores improved from pre-test to post-test for all measures; however, these differences were not statistically significant. When looking at subscale scores, the beliefs in treatment efficacy subscale of the PASAQ increased significantly (*p* = 0.03), and the readiness to screen subscale approached statistical significance (*p* = 0.06). Score changes were not significantly different for in-person versus virtual participants for any of the outcome measures.

In examining IAT group changes from pre-test to post-test, more participants were grouped in the ‘No preference’ and ‘White preference’ groups, post intervention. These differences approached statistical significance (*p* = 0.06, Figure 3).

## 4. Discussion

In this study, we examined the feasibility of implementing Black maternal mental health training for obstetric healthcare providers, comparing virtual and in-person training modalities. In this study, many outpatient obstetric care settings had significantly more MAs than RNs. This staffing imbalance may impact patient outcomes by limiting access to the specialized care that RNs provide, such as comprehensive assessments and in-depth patient education, both of which are critical for managing high-risk pregnancies and ensuring safe prenatal care [34]. The reliance on MAs, while efficient for routine tasks, may reduce the quality of personalized care, particularly for patients requiring more complex management [34]. This can potentially lead to missed opportunities for early detection of complications and inadequate support for patients with complex obstetric needs.

In this study, we observed trends towards improvement in most outcome measures, with statistically significant improvement in provider attitudes regarding the efficacy of PMAD screening. Provider readiness to screen and implicit bias change scores approached statistical significance. Because of the sampling techniques used and the voluntary nature of training participation, there was a ceiling effect in this study; that is, participants appeared to have positive views of PMAD screening and high levels of empathy at baseline. Thus, very small changes in outcome measures may not have been detected in this pilot sample. Findings at or near the level of significance for changes in readiness to screen, attitudes towards treatment efficacy, and implicit bias, post training, suggest that this program may enhance the ability of even well-informed providers to conduct PMAD screening and respond appropriately. Future studies should test this intervention with a broader sample and link training changes to health-system and patient outcomes, such as PMAD screening rates and referrals.

While the evidence of the impact of implicit bias training remains mixed, there is a moral obligation for healthcare systems and academic institutions to continue efforts to improve healthcare providers’ behaviors to enhance obstetric healthcare in the US [35]. The California Dignity in Pregnancy and Childbirth Act mandates that all clinicians providing perinatal care at hospitals, licensed alternative birth centers, or primary care clinics offering perinatal care services undergo evidence-based implicit bias training every two years [36]. This requirement should motivate these institutions to refine and perfect their programs, ensuring they achieve meaningful and lasting improvements in healthcare delivery. This training should incorporate real patient stories, the support of leadership, and a commitment to changing healthcare facility culture to sustain improvements in the quality of care [35]. Additionally, there must be transparency and a strong commitment to patients and the community, with healthcare staff held accountable for any discriminatory behavior or poor quality of care [37].

Addressing implicit bias among healthcare staff is essential for improving equitable care and reducing disparities in health outcomes. Implicit bias, particularly racial bias, can influence provider decision-making, affect patient–provider interactions, and contribute to unequal treatment. The findings of this study suggest that although the training increased participants’ readiness to screen for mental health issues and belief in treatment efficacy, it did not fully eliminate racial bias, specifically against Black individuals. This outcome is consistent with previous research, highlighting the persistent challenge of achieving long-term reductions in implicit bias among healthcare providers, indicating that brief interventions may not be sufficient to eradicate ingrained biases. Earlier studies involving obstetric medical residents and physicians indicated that while racial biases often remained, the training was successful in heightening awareness and motivating providers to implement strategies aimed at improving care quality [38]. Similarly, a study conducted with undergraduate students in Virginia and Canada demonstrated only a short-term reduction in racial bias, with no enduring impact on biases or behaviors [39]. Notably, the current study revealed a shift in racial preferences, with an increase in participants showing no preference between Black and White races, though it also uncovered an unexplained rise in preference for White over Black post training. These mixed results underscore the complexity of mitigating implicit bias.

One potential enhancement to improve the training module used in this study would be to incorporate more robust, continuous learning strategies. Instead of relying on a single training session, a longitudinal approach with follow-up modules, regular discussions, and periodic implicit bias assessments could help sustain changes in attitudes and behaviors. This continuous and on-going approach would allow for a deeper evaluation of long-term effectiveness and the persistence of attitude shifts over time. Moreover, expanding the scope of future studies to assess how the training addresses biases against other racial and ethnic groups—such as Asian, Hispanic, and Indigenous populations—would provide more comprehensive insights. Tailoring scenarios and case studies to reflect diverse patient identities could equip healthcare workers with a broader set of tools to effectively combat biases in various contexts, thus ensuring a more holistic approach to healthcare equity, beyond focusing solely on Black patients. Further steps to improve the study’s framework could involve developing adaptable, culturally sensitive modules that can be updated as new evidence and best practices emerge. This flexibility ensures that training remains relevant and impactful. Researchers should also prioritize collaboration with community representatives to ensure that diverse voices are integrated into study design and training content. Additionally, future research should measure long-term impacts on healthcare provider behavior and patient outcomes. Regular evaluation of these factors would allow for a more precise understanding of the training’s effectiveness and areas where it might fall short.

### Limitations

This study presents several strengths and limitations that are important for interpreting its findings. One notable strength is the relevance of the training content, which is grounded in the Reproductive Justice framework. This framework emphasizes the need for culturally relevant training that addresses the unique experiences of Black women, thereby enhancing the potential for equitable healthcare outcomes. Additionally, the inclusion of two cohorts—one participating in person and the other virtually—captures a wider range of experiences and learning preferences, increasing the generalizability of the results. Furthermore, the comprehensive content of the BMMH 101 training addresses critical gaps in knowledge and prepares healthcare providers to better serve Black women by covering topics such as implicit bias and mental health disparities in underserved populations.

However, the study has several limitations. One significant drawback is the self-selecting nature of participants; those who signed up for the study likely already possessed a passion for and awareness of the issues related to implicit bias. This predisposition may have influenced the mixed results observed in changes to attitudes and behaviors, as participants may have already been inclined to support equity in healthcare. Additionally, participants had considerable experience in their respective fields, averaging 24.9 years of experience. This level of experience may impact findings by limiting the potential for shifts in attitudes or practices, as more seasoned professionals might already have established beliefs and behaviors regarding patient care. Moreover, the training was offered only on weekends, which may have restricted participation to those who could accommodate such a schedule, potentially excluding individuals with conflicting obligations, such as work or family responsibilities. This could result in a sample that does not fully represent the broader population of healthcare providers. As a feasibility study, this research primarily aims to assess the practicality of implementing the BMMH 101 training, which may limit the generalizability of the findings. Therefore, future larger-scale studies should include randomized control designs and a more diverse sample of healthcare providers to ensure a more comprehensive understanding of the training’s efficacy and its broader implications for reducing implicit bias in clinical practice.

## 5. Conclusions

This study demonstrates that the BMMH 101 training can positively influence healthcare providers’ attitudes and knowledge regarding PMADs, with promising trends towards improving attitudes about treatment efficacy and readiness to screen. However, the persistence of some implicit racial biases and mixed outcomes reflects the complexity of addressing deeply ingrained biases within healthcare systems. The findings suggest that while short-term gains in knowledge and empathy are achievable, more robust and continuous education, incorporating real patient stories and ongoing reflection, is necessary to sustain long-term behavioral changes. Future research should focus on testing this intervention on a larger scale and linking it to patient outcomes, such as increased PMAD screening rates and better mental health support for minoritized communities. Healthcare institutions must also focus on fostering systemic changes that go beyond training to ensure meaningful, lasting improvements in perinatal care.

## Figures and Tables

**Figure 1 ijerph-21-01374-f001:**
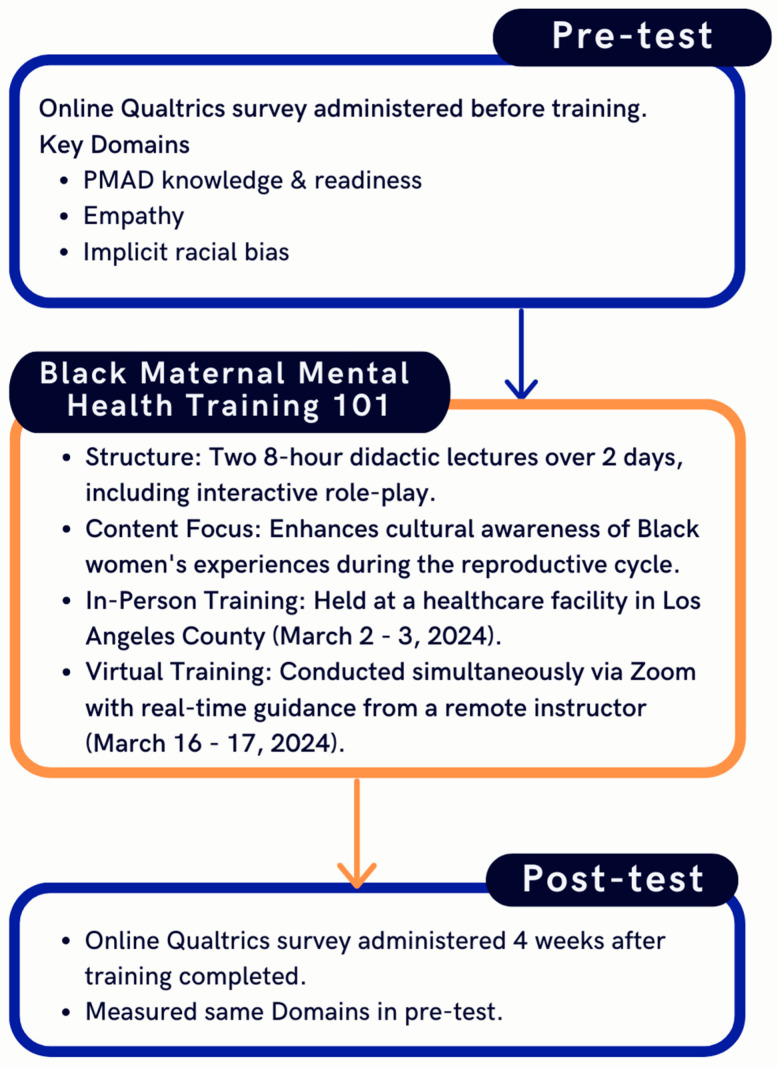
Study flow diagram.

**Figure 2 ijerph-21-01374-f002:**
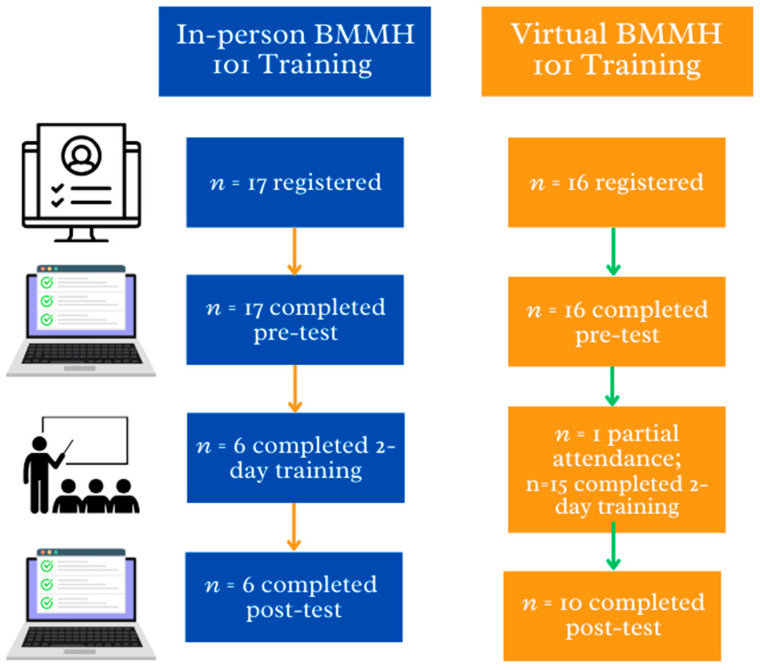
Participant recruitment flow diagram.

**Figure 3 ijerph-21-01374-f003:**
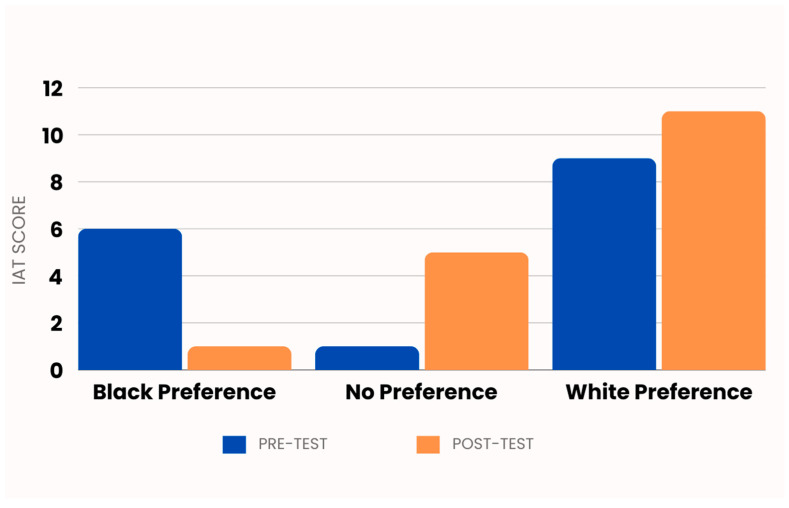
Pre-/post-test changes in implicit racial associations. Legend. This figure shows the frequency of responses from the Affect Misattribution Procedure implicit association test (IAT) according to racial preference in a sample of 16 obstetric healthcare providers before and after training on Black maternal mental health.

**Table 1 ijerph-21-01374-t001:** Black Maternal Mental Health 101 training content.

Session	Skills Acquired
Day 1: Major themes of (1) depression and anxiety disorders during pregnancy and after childbirth, (2) factors that affect risk of developing depression and anxiety disorders, (3) how to assess patients for depression and anxiety disorders.	Describe and identify signs/symptoms of depression and anxiety disorders. Understand clinician’s role in assessing and addressing depression and anxiety disorders. Roleplay and demonstrate proper assessment of depression and anxiety in patients. Review resources available to support patients.
Day 2: Major themes of (1) mental health in underserved populations, (2) implicit bias and empathy, (3) anti-racism in mental healthcare, (4) cultural and social influences on Black women’s mental health and receipt of care.	Understand the complex factors that influence availability and accessibility of mental health treatment for Black women. Describe implicit bias and its influence on clinician behaviors and treatment. Discuss and integrate empathy and cultural awareness into everyday life and clinical practice.

**Table 2 ijerph-21-01374-t002:** Measures.

Domain	Tool	Psychometric Properties	Sample Questions
PMAD knowledge and readiness to assess	24-item Perinatal Mood and Anxiety Disorders Attitudes and Screening Acceptability Questionnaire (PASAQ)	This tool was developed and psychometrically tested among pharmacists in Australia. The tool consists of 24 items on a 5-point Likert scale from 1 (strongly disagree) to 7 (strongly agree). The six components explored in the PASAQ include PMAD screening acceptability, screening readiness, stigma, attitudes towards treatment efficacy, medication counseling responsibility, and the effect PMADs have on others [31]. Internal consistency across subscales has been demonstrated (α 0.64–0.86). Because this study did not recruit pharmacists, the last four questions of the PASAQ were removed due to their focus on pharmacists’ role in PMAD screening and treatment.	Women with perinatal depression should not care for a child. Perinatal depression is common enough to warrant screening. I am confident in screening women for perinatal depression. I am likely to screen women for perinatal depression.
Empathy	20-item Jefferson Scale of Empathy (JSE): Health Professions version	This tool is a widely used instrument designed to assess the cognitive aspect of empathy, which involves patients’ experiences, perspectives, and feelings. The scale consists of 20 items, each rated on a 7-point Likert scale ranging from 1 (strongly disagree) to 7 (strongly agree). Total scores range from 20 to 140, with higher scores indicating higher levels of empathy. The tool has been validated in samples of nursing students and medical residents and has strong psychometric properties (α 0.81–0.85) [32].	My patients feel better when I understand their feelings. I try not to pay attention to my patients’ emotions in history-taking or in asking about their physical health. Asking patients what is happening in their personal lives is not helpful in understanding their physical conditions.
Implicit racial bias	The Affect Misattribution Procedure (Implicit association test, IAT)	The Affect Misattribution Procedure displays various visuals to participants on the computer screen to determine their differences in attitudes toward certain visuals. Some of the visual items are neutral figures, such as abstract paintings or Mandarin letters that are meant to prime participants. These neutral figures are then followed by a pictograph of, for the purposes of this study, a Black woman appearing distressed, then later a White woman appearing distressed. Participants then choose whether the visual is pleasant or unpleasant. This response is used to determine implicit race biases. The Affect Misattribution Procedure was developed in 2005, has since been used in multiple studies, and has strong predictive validity and reliability [33].	

**Table 3 ijerph-21-01374-t003:** Pre-/post-test differences in outcomes.

	Pre-Test		Post-Test		*p*-Value
	M	SD	M	SD	
PASAQ Total Score (20–100)	84.8	7.7	85.8	10.8	0.39
Acceptability of screening (2–10)	8.4	1.2	8.6	1.3	0.56
Stigma (7–35)	32.9	2.1	31.9	5	0.30
Attitudes towards treatment efficacy (3–15)	11.4	2.3	12.5	1.8	0.03
Readiness to screen (3–15)	11.8	2.5	12.5	2.3	0.06
Medication counseling responsibility (3–15)	11.3	2.2	11.2	1.9	0.81
Effects on others (2–10)	9.1	1.1	9.1	1.4	0.83
JSE Total Score (20–140)	119.6	11.6	118.9	15.2	0.81
Perspective taking (10–70)	61.7	6.5	61.2	6.9	0.67
Compassionate care (8–56)	44.9	5.7	41.1	9.0	0.94
Walking in patient’s shoes (2–14)	12.3	2.2	12.6	2.2	0.84

Notes: Pre-/post-test differences in outcomes in a sample of 17 obstetric healthcare providers after training on Black maternal mental health. M = mean; SD = standard deviation; PASAQ = Perinatal Mood and Anxiety Disorders Attitudes and Screening Acceptability Questionnaire; JSE = Jefferson Scale of Empathy.

## Data Availability

Data available upon request.

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
