# Peer review of "Enhancing Obstetric Healthcare Providers’ Knowledge of Black Maternal Mental Health: A Feasibility Study"

_ijerph, 2024, doi:10.3390/ijerph21101374_

Round 1

Reviewer 1 Report

Comments and Suggestions for Authors This study is a proof of evidence  to apply a model of training (BMMH101) for healthcare providers' attitudes and knowledge regarding PMADs, in order to avoid limitations and bias toward the black population.  The study has a complex protocol and looks well carried out. However the number of trainees is really modest and 70% were online and not in presence.  The results show that the racial preferences can be changed by the application of the training program in most of trainees and this is not seriously evaluated in the discussion. The implications of course are serious since this means that the training is per se biased toward a sort of racial discrimination.  This study can be worth publishing but authors should elaborate more than quoting just references 26 and 27 as potential solution. The training, according to this study, shows to be inadequate and should be challenged.  I would suggest a complete revision of the discussion and a more efficient statement toward the use of this training models before acceptance. 

Reviewer 2 Report

Comments and Suggestions for Authors

Thank you, authors, for this well-written article with clear arguments anchored in the background of the study. I have gone through it. However, there is a need to make a few things more clearer. My specific comments are here:

"Participants in both cohorts completed an online Qualtrics pre-test after reviewing study information."How was the pre-test conducted, and where and why? Are there any special reasons why the pre-test was conducted with the participants?

It is unclear how data for those who participated in person and those who participated virtually were analyzed. If they were both analyzed using R, version 4.3.1 then this should be made clearer.

Kindly provide a subsection describing the strengths and limitations of this article. Some of these are already in the discussion section. This will make this article unique and outstanding on Black maternal mental health issues.

I wish you all the best as you attend to these comments.

Reviewer 3 Report

Comments and Suggestions for Authors

Thank you for inviting me to review the paper by James et al, on 'Enhancing obesetrifc healthcare providers knowledge of black maternal mental health: a feasibility study'. The article covers an important topic and is of interest to clinicians. 

The article is very interesting and was written very clear. However, I have some minor comments just to improve the manuscript.

Methods:

Line 97 - Can you also include experience level of the person who has developed BMMH 101 training tool. 

Results:

I would appreciate if you could add a flow diagram of the study.

Table 3. what is M, mean ?. define it.

I can not find the results table of Line 160 - 161. Can you please include it. 

Discussion: Very nice

Clinical implications: I would suggest to add some more. 

If there any difficulties encountered during the study - it would be helpful to add in the manuscript. 

Regarding the sample selection - Can you please provide more details regarding how was the participants being selected and include in flow diagram. How many were contacted..and so. 

I am bit surprise all the included participants had an average of over 20 years of experience. Is there any specific reason to choose participants with heaps of experience ?

Do you think will the findings vary if you have had selected participants with smaller experience.

I hope these helps.

Overall, the manuscript is well written, enjoyed reading.
